# Physical Activity in Patients with Neuromuscular Disease Three Years after COVID-19, a Longitudinal Survey: The After-Effects of the Quarantine and the Benefits of a Return to a Healthier Life-Style

**DOI:** 10.3390/jcm13010265

**Published:** 2024-01-03

**Authors:** Ignazio Leale, Valerio Giustino, Paolo Trapani, Paolo Alonge, Nicasio Rini, Ivana Cutrò, Olga Leone, Angelo Torrente, Antonino Lupica, Antonio Palma, Michele Roccella, Filippo Brighina, Vincenzo Di Stefano, Giuseppe Battaglia

**Affiliations:** 1Sport and Exercise Sciences Research Unit, Department of Psychology, Educational Science and Human Movement, University of Palermo, 90133 Palermo, Italy; ignazio.leale@unipa.it (I.L.); valerio.giustino@unipa.it (V.G.); paolo.trapani02@community.unipa.it (P.T.); antonio.palma@unipa.it (A.P.); 2Department of Biomedicine, Neuroscience and Advanced Diagnostic (BIND), University of Palermo, 90129 Palermo, Italy; alongep95@gmail.com (P.A.); nicasio.rini@unipa.it (N.R.); ivana.cutro@community.unipa.it (I.C.); olga.leonee98@gmail.com (O.L.); angelo.torrente@unipa.it (A.T.); antlupica@gmail.com (A.L.); filippo.brighina@unipa.it (F.B.); 3Department of Psychology, Educational Science and Human Movement, University of Palermo, 90133 Palermo, Italy; michele.roccella@unipa.it

**Keywords:** COVID-19, coronavirus, pandemic, quarantine, lockdown, physical activity, exercise, training, home exercise, neuromuscular diseases

## Abstract

Background: Quarantine was one of the strategies adopted by governments against the spread of COVID-19. This restriction has caused an increase in sedentary behaviors and a decrease in the practice of physical activity (PA), with a consequent negative impact on lifestyle both in healthy people and in those who need constant practice of PA to combat diseases, such as patients suffering from neuromuscular diseases (NMDs). Hence, this study aimed to compare PA levels among patients with NMD during and after quarantine. Methods: An adapted version of the International Physical Activity Questionnaire Short-Form and the Short-Form Health Survey were administered during COVID-19 quarantine (T0) and after 3 years (T1) to 91 Italian patients with NMDs. Results: We found a significant increase in the total PA level at T1, with no significant changes in vigorous-intensity PA. Moreover, a significant decrease in the PA level was found among the patients with different NMDs. No significant changes in physical component scores and mental component scores were detected. Conclusions: Our results suggest that it would be necessary to provide alternative indoor exercise settings to prevent the adoption of sedentary behaviors.

## 1. Introduction

The COVID-19 pandemic, which lasted from late 2019 to 2022, significantly changed the lives of millions of people worldwide. Indeed, various government restrictions have led to an increase in sedentary behaviors and a decrease in the practice of PA. The main restrictive measures included quarantine, social distancing, and travel blocking, with numerous negative effects on the global population including healthy people and patients. The restrictive measures adopted to contain the spread of COVID-19, with a consequent decrease in PA levels compared to the pre-pandemic condition and an increase in sedentary behaviors, have led to negative effects both in healthy people and patients suffering from various diseases, such as patients suffering from NMD [1,2,3,4,5]. 

NMDs are heterogeneous conditions that affect both children and adults to varying degrees between patients causing relevant disability [6,7]. NMDs include disorders of the musculoskeletal system (e.g., muscle dystrophies, a disease that causes the degeneration of skeletal muscles), neuromuscular junction (e.g., myasthenia gravis, an autoimmune disorder with fluctuating weakness in skeletal muscles and specific autoantibodies of the acetylcholine receptor, or less frequently against muscle-specific kinase at the neuromuscular junction), motor neurons (e.g., amyotrophic lateral sclerosis, a neurodegenerative disease that causes progressive destruction of motor neurons), and peripheral nerves (e.g., familial amyloid neuropathies, chronic inflammatory demyelinating polyradiculoneuropathy). Charcot–Marie–Tooth disease is the most common hereditary peripheral disease, and it is estimated that 1 person in 2500 is affected by this pathology [8]. 

The causes of NMDs are different; some are hereditary, and others are acquired and developed throughout life. Hereditary diseases are caused by genetic mutations, present at birth, while acquired diseases may depend on environmental factors or factors such as infections, inflammation, and trauma. 

Unfortunately, not all neuromuscular diseases have a cure; sometimes the only purpose of treatment is to improve the quality of life of patients and reduce the symptoms as much as possible [9]. The common symptoms include fatigue, loss of muscle mass, balance problems, cramps, and tremors [10,11].

Patients with NMDs require immunotherapies, muscular rehabilitation, exercise, and strict follow-up [6,7,12,13,14,15]. Exercise and PA are fundamental in NMDs to improve muscle strength and endurance and prevent musculoskeletal complications resulting from disuse [7,12,13,14,16]. Furthermore, it should be noted that exercise and PA should be carried out by specialists in sport and exercise sciences and in collaboration with physiotherapists in order to ensure a specific exercise protocol, individualized and adapted to the pathology, as well as the needs of each patient [7,12,13,14,16]. 

During the COVID-19 pandemic, due to the government’s safety measures, Italian NMD patients changed their daily habits and lifestyle. As a matter of fact, several studies showed that the PA levels decreased during this period both in healthy and people with NMDs of all ages [17,18,19,20,21,22,23,24,25,26,27,28,29]. The decrease in PA practice has been demonstrated to have a negative impact on the mental and physical sphere and disease symptoms due to the difficult access to treatments [1,7]. Of note, the World Health Organization recommended continuing the practice of PA during the lockdown (i.e., at least 150 min/week of moderate-intensity PA, or 75 min/week of high-intensity PA [30]) to counteract these negative effects [31]. Some home-based exercises suggested include online exercise, jumping rope, dancing, active video gaming, and balance exercises which can be easily done at home [32]. 

A previous study by our research group showed the negative impact of COVID-19 quarantine on PA levels in patients with NMDs [28]. In that study, we investigated the level of PA through an adapted version of the International Physical Activity Questionnaire Short-Form, detecting a significant reduction in the total weekly PA level, and in particular, in walking activities among patients with NMDs. Moreover, we found a correlation between the magnitude of PA reduction and the physical and mental health component scores of the Short-Form Health Survey. Furthermore, a correlation between PA levels and SF-12 scores was demonstrated. In particular, neurodegenerative disorders showed lower physical health component scores (PCS-12 scores) compared to patients affected by myopathy, and lower mental health component scores (MCS-12 scores) compared to patients affected by polyneuropathy and myasthenia. More recently, in the years after the pandemic, patients affected by NMDs have gradually conducted a full return to normal and routine care. They have started exercise, physiotherapy, and PA. However, there is no evidence for a return to pre-pandemic levels. Indeed, as NMDs are neurodegenerative and progressive disorders, an incomplete recovery is expected due to prolonged immobilization and sedentary behaviors. However, it is also true that the recovery may depend on the length of immobilization and on the extent of PA reduction.

Therefore, based on those findings, in the present follow-up study, we aimed to compare the level of PA during the COVID-19 quarantine and three years after the pandemic, i.e., after returning to pre-pandemic levels, in patients with NMDs.

## 2. Methods

### 2.1. Study Design

This is a follow-up study in which a survey, conducted via telephone interview, was administered. The protocol used was similar to our previous study published in 2020 [28]. The survey included an adapted version of the International Physical Activity Questionnaire Short-Form (IPAQ-SF) [17,28] and a Short-Form Health Survey (SF-12) [33]. 

Each participant was interviewed during COVID-19 quarantine (T0) and three years after the pandemic (T1), and the scores between T0 and T1 were compared.

To be included in the research protocol, all participants signed informed consent. The Bioethics Committee of the University of Palermo approved the study in accordance with the principles of the Declaration of Helsinki.

### 2.2. Participants

Participants were recruited between 20 April and 4 May 2020 (i.e., during the COVID-19 quarantine in Italy) at the Neuromuscular Clinic of the University of Palermo and were followed until June 2023 (i.e., up to three years after the start of the pandemic). 

The inclusion criteria were diagnosis of NMD among the following: acquired or hereditary myopathy (MY); acquired or hereditary polyneuropathy (PN); neuromuscular junction disorder (NJ); genetically confirmed degenerative disease (ND, i.e., hereditary spastic paraplegia, spinal muscular amyotrophy). The exclusion criteria were as follows: (1) lack of follow-up at 3 years; (2) drop out of the study.

### 2.3. Measurements

The survey was administered, via telephone interview, by the same researcher during the COVID-19 quarantine (T0) and three years after the pandemic (T1) and included an adapted version of the IPAQ-SF and the SF-12.

The first phase of the telephone interview involved a careful explanation of the questionnaires, analyzing each item, in order to reduce the possible errors in understanding each participant. The questionnaires were administered in a specific order: the first questionnaire administered was the IPAQ-SF followed by the SF-12. A Microsoft Excel spreadsheet (Microsoft Corp, Redmond, WA, USA) was created in order to collect information related to the items of IPAQ-SF and SF-12, and to ensure a correct data collection process.

The adapted version of the IPAQ-SF measured the PA practice of the previous 7 days, both at T0 and T1, in terms of energy expenditure (MET—minutes/week) [17,28]. In detail, this tool analyzes the types, frequencies, and durations of each PA intensity (i.e., walking activities, moderate-intensity PAs, and vigorous-intensity PAs) [17]. The questionnaire distinguishes PA into moderate PA and vigorous PA. Moderate-intensity activity is considered a physical effort that forces you to breathe at a higher rate than normal, while vigorous-intensity activity is considered a physical effort that does not allow the subject to speak.

SF-12 is a tool composed of 12 items (derived from the original questionnaire SF-36) that allows for the measurement of two specific domains: PCS and MCS [33]. In detail, the PCS domain consists of six questions of which two are related to PA, two concern the limitations and physical health caused by diseases, one concerns physical pain, and one is related to the perception of the individual about the proper state of health. The MCS domain consists of six questions, of which two are related to the emotional state, one concerns vitality, one concerns social activity and two refer to the mental health of the subject. These items are used because they allow us to evaluate the physical functioning, functional limitations, body pain, the perception of general health, social functioning, and the mental and physical state of the participants. Overall, it allows us to evaluate the correlation between the quality of life and health in the target population.

### 2.4. Scoring Protocol

Regarding the adapted version of the IPAQ-SF, we analyzed the walking activities, moderate-intensity PAs, vigorous-intensity PAs, moderate-to-vigorous PAs (MVPAs), and total PA levels for both T0 and T1. 

The PA levels were measured as energy expenditure in MET—minutes/week using the metabolic equivalent task (MET) known for each PA intensity (i.e., 3.3 for walking activities; 4.0 for moderate-intensity PAs; 8.0 for vigorous-intensity PAs) [34]. The sum of MET—minutes/week of moderate-intensity PAs and vigorous-intensity PAs was calculated for the energy expenditure of the MVPAs and the sum of MET—minutes/week of all the PA intensities was calculated for the total PA level [17,35]. The scoring was computed by multiplying the MET of each PA intensity per the amount of practice known (i.e., minutes) during the previous 7 days (i.e., MET—minutes/week) (http://www.ipaq.ki.se accessed on 29 October 2023) [17,35,36,37]. The total MET allows for the allocation of each subject within a reference category: if the total is less than 700 MET, the subject is inactive; if the total is between 700 and 2519 the subject is sufficiently active, if the total is above 2520 the subject is active or very active.

Regarding SF-12, the scores obtained in the two domains of PCS and MCS were calculated which, as is known, range from 0 to 100, with higher scores indicating a better health-related quality of life [33,38,39,40]. 

### 2.5. Statistical Analysis

The categorical variables were reported as percentages, and continuous variables as the means ± standard deviations. The Chi-square test was used to compare categorical variables between groups, and the Mann–Whitney U test to compare continuous variables (i.e., total weekly PA level at T0 and T1). Boxplots/histograms were used to represent the variables. 

All the statistical analyses were performed using SPSS Statistic (version 26.0 IBM Statistics, IBM Corp. Lane Cove, NSW, Australia) with a level of significance set at <0.05. Graphs were created using GraphPad Prism 7 (GraphPad Software Inc., San Diego, CA, USA).

## 3. Results

### 3.1. Participants

At T0, 149 patients with NMDs were included in the study as published in our previous study [28]. These patients should have been followed for three years and evaluated at the end. However, 58 patients were not evaluated at T1 (26% were not traceable and had never presented for a check-up in the three years; 5% dropped out of the study; and 3% were not evaluable due to death). Hence, 91 Italian patients with NMDs completed the study participation. Among these, 49 were males (54%) and 42 were females (46%). Overall, 82 patients (90%) were able to walk independently, while 9 patients (10%) had a walking impairment. The NMDs were distributed as follows: 11 patients had an acquired or hereditary myopathy (MY); 46 patients had an acquired or hereditary polyneuropathy (PN); 31 patients had neuromuscular junction disorder (NJ); while 3 patients suffered from hereditary spastic paraplegia (HSP), as shown in Figure 1. 

In the survey at T1, 15 patients (17%) stated an improvement in the disease, 41 patients (45%) stated a stability in the disease, and 35 patients (39%) reported a worsening of the disease.

No significant differences (*p* = 0.579) in BMI values were found between T0 and T1.

### 3.2. The Adapted Version of the IPAQ-SF and SF-12

A descriptive analysis of the levels of MET—minutes/week for each PA intensity and the total weekly level of PA at T0 and T1 was calculated. Table 1 reports these parameters at T0 and T1, and the differences between them.

The Mann–Whitney U test showed a significant difference in MET—minutes/week of walking PAs (*p* < 0.001), moderate-intensity PAs (0.003), MVPAs (0.002); and in the total PA (*p* < 0.001). No significant difference was detected in the MET—minutes/week of vigorous-intensity PAs between T0 and T1. 

Of interest, MET—minutes/week of walking PAs and total PA increased at T1 in MY and NJ patients with a low level of improvement in PN and HSP patients (Figure 2). 

Regarding walking, no significant differences were found at the baseline, but walking participants presented a better recovery at T1 compared to wheelchair patients who presented reduced PA levels (MET tot 961 ± 378.9 vs. 120 ± 413.6 min/week, *p* = 0.004). Indeed, MET walking was the most significantly improved parameter after the pandemic, but it may not improve in patients with impaired walking. Moreover, as shown in Figure 3, the Mann–Whitney U test showed no significant changes in both SF-12 domains, i.e., PCS-12 (*p* = 0.681) and MCS-12 (*p* = 0.512).

## 4. Discussion

This is a follow-up study of that published in 2020, in which we aimed to measure the levels of PA in a sample of 268 Italian patients with NMDs before and during the last week of quarantine, finding a significant decrease in PA levels during COVID-19 quarantine [28]. The aim of the present study was to compare the level of PA during the COVID-19 quarantine and three years after the pandemic, i.e., after the return to normal life, in patients with NMDs. 

Pandemic restrictions related to COVID-19 have led to direct and indirect health implications. Quarantine appears to have had a negative impact on some characteristics related to the quality of life of both people with disabilities and the healthy population. In fact, the lack of or reduction in access to specialist care and facilities has played a fundamental role in this phenomenon. In our previous 2020 study, we reported a reduction in PA levels and perceived mental and physical health in both NMD patients and healthy controls during the pandemic [28]. Other studies reported similar findings in patients with NMDs [1,41] and patients with other neurological conditions [42,43]. Of interest, a study conducted in children and young adults affected by intellectual/physical disabilities showed the negative effects of quarantine on PA and investigated the long-term consequences of quarantine restrictions [44]. In contrast, research showed that 31% of patients with multiple sclerosis inactive during the pandemic had no plans to change their PA practice once restrictions were lifted (42% were unsure) [45]. 

To our knowledge, there are no studies that have investigated whether the amount of PA practice returned to pre-pandemic levels after the removal of COVID-19 restrictions. Therefore, no light has been shed on the progressive return to PA and/or the difference between PA levels during and after quarantine, both in the general population [46] and patients with NMDs.

The present study showed that, three years after the pandemic, the MET—minutes/week of walking PAs, moderate-intensity PAs, MVPAs, and total PA increased significantly. Moreover, the maximum increase was recorded for walking MET, especially for the subgroups of MY and NJ (Figure 2), thus confirming that walking, as shown in other populations [47], is a good practice also for people with NMDs, promoting a healthy lifestyle. Of note, in PN patients, MET walking did not significantly improve, probably because they are often in a wheelchair (Figure 2). Hence, patients with good walking ability could experience better recovery, in contrast with patients with a wheelchair who were more severely affected by restrictions. In these fragile patients, prolonged immobilization might result in an abrupt interruption of protein synthesis in skeleton muscles, thus causing muscle mass loss. In this perspective, a different approach should be proposed to improve lifestyle in patients with impaired ambulation (i.e., exercise with the upper limbs). Of interest, in patients with MY, the levels of PA were even higher than in the pre-pandemic period, while PN patients did not return to previous levels. 

However, the perceived mental and physical health, measured via the SF-12, did not differ from the pandemic period. Similar results were found for the BMI. These results should be compared with the general population. Although several studies have highlighted the effects of isolation on weight, anxiety, depression, and PA management as a short-term effect of the COVID-19 pandemic, no follow-up studies have been conducted to date to evaluate the impact of the gradual return to healthier habits [48]. Our previous article highlighted how, compared to the general population, SF-12 scores were lower in patients with NMDs during quarantine. Even after three years, patients did not achieve a better quality of life, regardless of resuming PA. Nearly a third of them reported experiencing a worsening of their disease, and only 14% reported improvement.

However, it should be considered that 51% of patients in this cohort were affected by chronic and progressive conditions such as polyneuropathies (Figure 1). We suppose that in such chronic and progressive disease subgroups, a complete return to pre-pandemic PA and disability levels was not possible.

These findings suggest that the abrupt interruption of usual physical activities during the pandemic might have favored a worsening of such subtypes of NMD; among disease groups, PN and ND patients might be more prone to clinical progression compared to autoimmune conditions like NJ and CIDP that are treatable with effective drugs that reduce disease progression [49]. Considering the progressive nature of the former conditions, the restoration of previous PA levels may be difficult. Hence, indoor exercise programs should be encouraged in the case of patients’ home confinement. In this context, tele-coaching and telemedicine might represent a unique opportunity to preserve muscle mass by ensuring continuous training and an adequate lifestyle.

Finally, it is also important to underline that lifestyle impacts anthropometric measures as well. Indeed, increased MET after three years was also associated with a slightly improved BMI, even if not significant (Table 1). This result confirms the strict correlation between PA and BMI that should be considered in NMD patients, who are more often prone to obesity and cardio-respiratory events [50].

In light of these findings, it might be useful to implement an active lifestyle culture, for example through the execution of a training protocol in tele-coaching. Indeed, previous studies have shown that tele-coaching can be used efficiently to support intervention programs such as the cardiopulmonary resuscitation and positive pressure ventilation [51]. Tele-coaching is a new training method that involves the use of information technologies and digital tools, such as computers, tablets, and phones in order to access a training protocol remotely. In particular, tele-coaching training involves the use of demonstration videos, the production of a training manual with related references to the training protocol, a training diary in which to monitor adherence to the protocol, as well as specialized coaches available to clarify any doubts of the patient. This type of training has already been developed in other populations, resulting in effective and risk-free injury [52,53]. However, few studies have been performed on patients with neuromuscular diseases [54,55]. Chetlin et al. (2004) used a home-based resistance training program for 12 weeks, with the aim of evaluating the effectiveness of the protocol in tele-coaching for body composition, strength, and functionality in quotidian activities in patients suffering from Charcot–Marie–Tooth [54]. El Mhandi et al. (2008) used a training approach in tele-coaching to evaluate cardiorespiratory, neuromuscular, and functional abilities, as well as the perception of fatigue and pain in patients with Charcot–Marie–Tooth [55]. The results showed significant improvements in cardiorespiratory ability, functional abilities, and reduction in pain and fatigue. These studies demonstrate the efficiency and feasibility of this new training modality in this target population, allowing to increase levels and time devoted to the practice of physical activity. Overall, it might be useful to extend the administration of this mode of physical activity in this population in order to break down the barriers related to costs, travel, and organizational difficulties.

## 5. Conclusions

In conclusion, our study showed that quarantine negatively affected PA practice habits and lifestyle in patients with NMDs. These findings are in line with the existing literature showing a decrease in PA and an increase in sedentary behaviors after the pandemic. These changes have left their mark on patients as they report a lower level of the quality of life compared to the pre-pandemic period and a worsening of the disease even after three years. These results show that governments should choose containment strategies very carefully and avoid quarantine, if possible. It has a negative impact not only on physical activity, but also on the health of subjects.

Further investigations are needed in order to compare these results with those obtained from the general population and to clarify whether psychological or physical factors related to the disease may have influenced the return to pre-pandemic habits.

### 5.1. Strengths and Limitations

The study was conducted at a follow-up, three years after the start of the pandemic, using the same assessment questionnaires, which were administered by the same previous researchers. Moreover, the results of our study suggest the need to increase physical activity levels in this population. This consideration highlights the possibility of applying a new training method, i.e., tele-coaching, that is an effective, safe, and risk-free type of training. The assessment of PA and quality of life levels through questionnaires administered is one of the major limitations of this study because administration of the questionnaires via telephone calls causes a high loss of follow-up patients due to non-response to the call. Finally, a limitation comes from the high rate of drop out. This could be explained in part by mortality from COVID-19 in NMD patients, as reported, for example, in NJ patients [56]. However, a significant number of patients opted not to participate in the follow-up survey. More studies are still needed to confirm these results and explore the long-term consequences of the pandemic on lifestyle habits in NMD patients.

### 5.2. Practical Implications

Therefore, our study highlights the need for governments to be very careful to re-insert a quarantine because it has a negative impact not only on the practice of physical activity, but also on health. Moreover, based on our findings, this study highlights the need to implement a culture of alternative PA practice interventions, such as tele-coaching, to preserve the habit of PA even under adverse conditions, and improve the quality of life in this population. These alternative PA practice interventions are aimed at specialists in sport and exercise sciences who deal with physical exercise to whom we recommend applying them in cases where physical exercise is essential to combat diseases, such as in patients suffering from NMDs. This can ensure a specific exercise protocol, individualized and adapted to the pathology, as well as the needs of each patient.

## Figures and Tables

**Figure 1 jcm-13-00265-f001:**
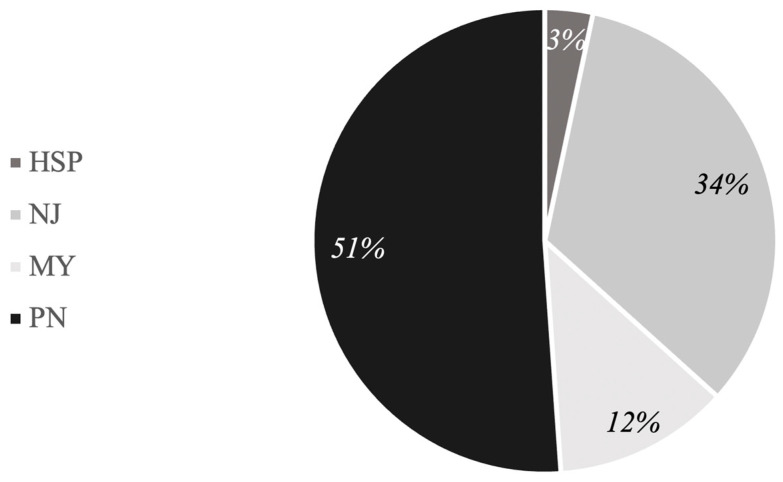
Percentage distribution of the NMDs. MY, acquired or hereditary myopathy; PN, acquired or hereditary polyneuropathy; NJ, neuromuscular junction disorder; HSP, hereditary spastic paraplegia.

**Figure 2 jcm-13-00265-f002:**
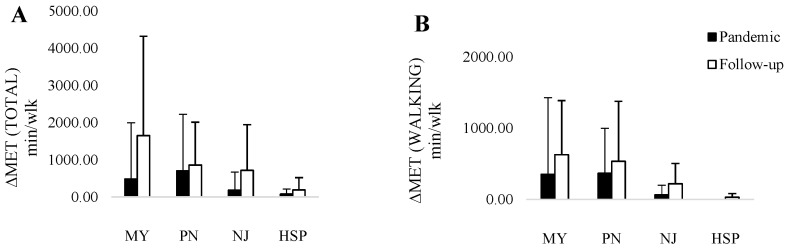
MET—minutes/week of total PA (**A**) and walking PAs (**B**) according to patients’ NMDs. MY, acquired or hereditary myopathy; PN, acquired or hereditary polyneuropathy; NJ, neuromuscular junction disorder; HSP, hereditary spastic paraplegia.

**Figure 3 jcm-13-00265-f003:**
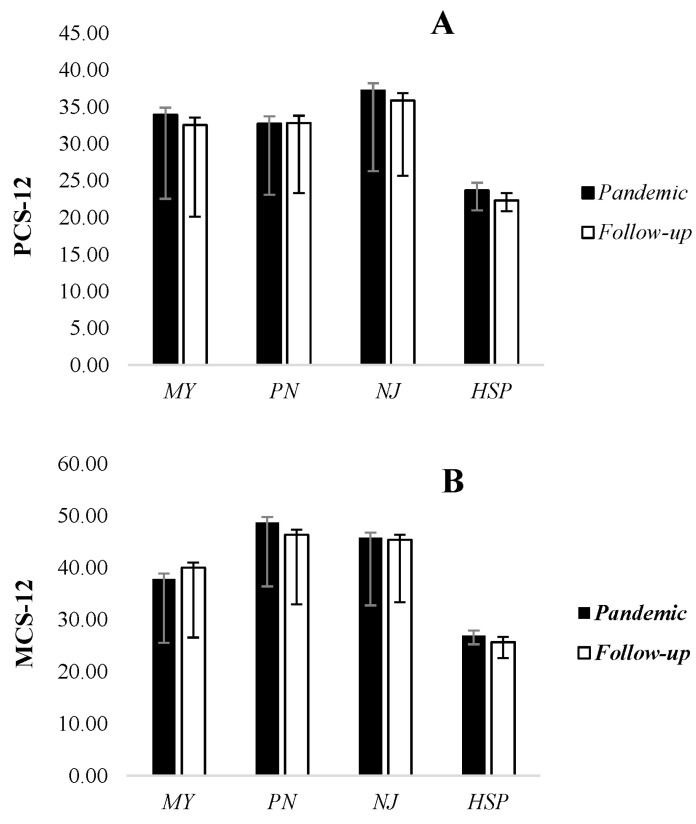
PCS-12, physical health component score of the SF-12; (**A**) MCS-12, mental health component score of the SF-12 (**B**). MY, acquired or hereditary myopathy; PN, acquired or hereditary polyneuropathy; NJ, neuromuscular junction disorder; HSP, hereditary spastic paraplegia.

**Table 1 jcm-13-00265-t001:** Demographics data and PA levels of the entire sample at T0 and T1 with the differences analyzed using the Mann–Whitney U test.

	During the Pandemic (T0)	Follow Up (T1)	*p*-Value
Age (years)	54.98 ± 13.437	57.65 ± 12.859	0.212
Gender (males)	49/91 (53.8%)	49/91 (53.8%)	1
BMI (kg/m^2^)	26.9 ± 4.9	23.87 ± 4.98	0.579
MET—minutes/week of walking PAs	252.03 ± 597.1	656.7 ± 2324.75	<0.001 ***
MET—minutes/week of moderate-intensity PAs	174.5 ± 538.9	1747.69 ± 13,570.4	0.003 **
MET—minutes/week of vigorous-intensity PAs	52.75 ± 381.6	134.51 ± 705.8	0.251
MET—minutes/week of MVPAs	227.25 ± 817.54	1882.2 ± 13,581.2	0.002 **
MET—minutes/week of total PA	479.29 ± 1244.76	2538.9 ± 15,821.7	<0.001 ***

**, *p* < 0.01; ***, *p* < 0.001.

## Data Availability

Data are available from the corresponding author upon reasonable request.

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
