# Peer review of "Physical Activity in Patients with Neuromuscular Disease Three Years after COVID-19, a Longitudinal Survey: The After-Effects of the Quarantine and the Benefits of a Return to a Healthier Life-Style"

_jcm, 2024, doi:10.3390/jcm13010265_

Round 1

Reviewer 1 Report

Comments and Suggestions for Authors

JCM-2665549 – Physical activity in patients with neuromuscular disease 3 years after COVID-19

This is an interesting, clinically relevant, and topical longitudinal survey that sought to evaluate the effects of COVID-19-imposed quarantine on physical activity levels of 91 patients with neuromuscular disease.

The survey is performed reasonably well, and the findings are worthy of investigation. More background and reasons for selection of the 2 instruments (IPAQ-SF and SF-12) is suggested, including prior use of these questionnaires in similar settings. The introduction is unnecessarily long, and the causes, symptoms, and treatments of neuromuscular diseases do not relate to the objective of the report.  On the other hand, a description of their survey tools (IPAQ-SF and SF-12 questionnaires) would be very helpful for the reader who is not facile with their application, validity, and limitations. Most importantly is the need for the authors to explain to the reader how the changes in the reported questionnaires support some of the authors’ (reasonable and probably helpful) recommendations.

Specific comments:

L: line number

The title should indicate the type of article (perhaps longitudinal survey).

The manuscript is unnecessarily difficult to read because of the excess of abbreviations – there are, for instance, 7 abbreviations in the abstract alone. 

L42 – NMD was already defined in L41 – it is redundant.

L45-6 – This is a medical journal – there is no need to define “muscle dystrophies,” – and certainly not “skeletal muscles.”

L47 – The reviewer disagrees that myasthenia is a “malfunction of the immune system” – please be precise.

L49 – No need to define “motor neurons.”

L92 – How are “exercise” and “PA” different?

L96 – What does “the entity of PA reduction” mean? Should it be “extent of PA reduction”?

L99 – What do the authors mean by “normal life”? Is it “pre-pandemic”?

L105 – What do the authors mean by an “adapted version” of the IPAQ-SF? Was this adapted version validated? 

L144-5 – The description is redundant (see L142).

L146 – To what “role” do the authors refer?

L275 – Please define MY and NJ before abbreviating.

L275-6 – Please reference this statement.

L277 – Do the authors mean, “ambulatory”?

L278 – Suggest, “in contrast” rather than “at difference.”

L280 – Suggest, Result” instead of “conduct.”

L323 – “Effective and risk-free” what? Please complete the sentence.

L351 – Please elaborate on how the authors introduced this “new training method.”

L356-7 – Were follow-up calls attempted if the initial call was not answered?

L365-6 – How do the authors’ findings support the need to implement telecoaching, for instance?

Comments on the Quality of English Language

Overall, the authors did a reasonable English presentation. The reviewer attempted to suggest some changes that would improve the quality, but further editing for syntax would be helpful.

Author Response

Dear Editors and Reviewers,

Your comments on our manuscript are much appreciated. Our Manuscript was improved thanks to the valuable and helpful comments of the Reviewers and Editors. The comments were carefully examined and corrections were made. For this reason, we are submitting our manuscript again for your kind consideration for possible publication.

Point-by-point reply.

Reviewer #1:

  • The title should indicate the type of article (perhaps longitudinal survey).

R: Thank you for this suggestion. We changed the title to " Physical activity in patients with neuromuscular disease three years after COVID-19, a longitudinal survey: the after-effects of the quarantine and the benefits of a return to a healthier lifestyle”.

  • The manuscript is unnecessarily difficult to read because of the excess of abbreviations – there are, for instance, 7 abbreviations in the abstract alone.

R: Thank you for this suggestion. We revised the entire manuscript and reduced the abbreviations in the abstract and other sections.

  • L42 – NMD was already defined in L41 – it is redundant.

R: Thank you for the correction. The term has been corrected.

  • L45-6 – This is a medical journal – there is no need to define “muscle dystrophies,” – and certainly not “skeletal muscles.”

R: Thank you for the suggestion. We revised the manuscript according to the indications.

  • L47 – The reviewer disagrees that myasthenia is a “malfunction of the immune system” – please be precise.

R: We thank the reviewer for this punctual suggestion. We replaced the definition with “myasthenia gravis, an autoimmune disorder with fluctuating weakness in the skeletal muscles and specific autoantibodies towards the acetylcholine receptor, or less frequently against muscle specific kinase at the neuromuscular junction”

  • L49 – No need to define “motor neurons.”

R: Thank you for the suggestion. We changed the manuscript according to the indications.

  • L92 – How are “exercise” and “PA” different?

R: The World Health Organization defines motor activity as any bodily movement produced by our body that requires energy expenditure. The term physical exercise, on the other hand, means a structured activity, planned and performed regularly.

  • L96 – What does “the entity of PA reduction” mean? Should it be “extent of PA reduction”?

R: Thank you for the comment. We have corrected the manuscript according to the indications.

  • L99 – What do the authors mean by “normal life”? Is it “pre-pandemic”?

R: Thank you for the suggestion. This term has been corrected.

  • L105 – What do the authors mean by an “adapted version” of the IPAQ-SF? Was this adapted version validated?

R: Thank you for this comment. The term “Adapted version” of the questionnaire indicates a reduced version of the original SF-36, that is the SF-12.

  • L144-5 – The description is redundant (see L142).

R: Thank you for this comment. The period has been corrected and made clearer.

  • L146 – To what “role” do the authors refer?

R: Thank you for the comment. The concept has been revised and corrected.

  • L275 – Please define MY and NJ before abbreviating.

R: Thank you for the comment. The terms have been defined previously L-199-201

  • L275-6 – Please reference this statement.

R: Thank you for the comment. The reference has been inserted.

  • L277 – Do the authors mean, “ambulatory”?

R: Thank you for the comment. The concept has been cleared in the manuscript.

  • L278 – Suggest, “in contrast” rather than “at difference.”

R: Thank you for the comment. The term has been changed in the manuscript according to the indications.

  • L280 – Suggest, Result” instead of “conduct.”

R: Thank you for the comment. The term has been changed in the manuscript according to the indications.

  • L323 – “Effective and risk-free” what? Please complete the sentence.

R: Thank you for the comment. The concept refers to the lack of risk for injury. The concept has been clarified in the manuscript.

  • L351 – Please elaborate on how the authors introduced this “new training method.”

R: This type of training is defined as a "new mode of training" because, although much used for workouts in healthy people, it is rarely applied in patients with pathologies. However, this mode could help patients with neuromuscular diseases, breaking down numerous barriers (travel, costs, training facilities…).

  • L356-7 – Were follow-up calls attempted if the initial call was not answered?

R: Thank you for the comment. We performed multiple calls over the days. After a week without a response, the patient was excluded.

  • L365-6 – How do the authors’ findings support the need to implement telecoaching, for instance?

R: Thank you for the comment. The use of telecoaching can be one of the main strategies to be used to increase physical activity levels in these patients. Telecoaching allows you to perform training in your home environments, at the preferred times and days, as well as reduce the need to use specialized facilities. Therefore, this training modality reduces barriers that do not allow the execution of activities in patients with neuromuscular diseases.

Hoping in a positive feedback we look forward to hearing from you soon.

Kind regards,

Vincenzo Di Stefano

Reviewer 2 Report

Comments and Suggestions for Authors

In this paper it was found that the level of physical activity in patients with neuromuscular disorders was higher 3 years after COVID than during COVID-19 quarantine. This suggested that quarantine reduced the levels of physical activity and it was concluded that ‘it would be necessary to provide alternative indoor exercise settings to prevent adoption of sedentary behaviors’.

The title needs to change, as the impression is easily given by the title that the observed reduction in physical activity are a consequence of the pandemic and COVID, where really it is a consequence not of the pandemic or COVID, but rather a consequence of quarantine measures! I therefore suggest something along the lines of the following:

‘Reductions in physical activity level in patients with NMD is a consequence of quarantine during COVID’

The second issue is the conclusion: I would conclude that

‘Physical activity levels had returned to normal three years after quarantine, but governments should be careful in implementing quarantine as it has a negative impact on physical activity that in turn has negative consequences for health’.

I rather think these messages should be send out, as the way COVID was dealt was highly inappropriate - that is my opinion - but irrespective of my opinion your findings indicate that those quarantine measures had bad consequences for physical activity and that should be the conclusion.

In the abstract the abbreviations PCS and MCS are used, but they are not explained in the abstract.

Of the 9 patients with walking impairment, did they also have a walking impairment at baseline?

On page 12 you use the word ‘difference’, but where possible, please rather tell whether it was increased or decreased.

Is the y-axis in figure 2 really ‘delta MET’? I think it is just ‘MET’.

Discussion

Page 7 line 275: I am not sure that ‘the maximum increase in walking MET confirms the role of daily walking for a healthier lifestyle’ It is PART of that lifestyle, but not confirmation. It reads like: ‘’Walking is good for you. More walking we have seen. This confirms walking is good for you’.

It seems you have also pre-pandemic data. Would it not be interesting then to see if the PA levels, PCS and MSC etc had returned to pre-pandemic levels, or remained lower?

Page 9 line 351-352: Your study did NOT introduce a new training method. You perhaps discussed it in the Discussion, but you did not introduce it in the Methods, or described any results of the training in the Results section. This, therefore, is an inappropriate comment that should be removed.

Page 9 line 358: Mortality was NOT a major issue in your study. It was just 3% (and perhaps an expected percentage for these conditions in 3 years. A larger problem was drop out (5%) and even more a lack of traceability (26%) (see page 4 for these numbers).

The implication is that one should be extremely hesitant to introduce another lock-down/quarantine period!

Comments on the Quality of English Language

Dear Sir/madam,

I read the paper with interest and it reads well. I have some minor suggestion.

Author Response

Dear Editors and Reviewers,

Your comments on our manuscript are much appreciated. Our Manuscript was improved thanks to the valuable and helpful comments of the Reviewers and Editors. The comments were carefully examined and corrections were made. For this reason, we are submitting our manuscript again for your kind consideration for possible publication.

Point-by-point reply.

Reviewer #2:

  • The title needs to change, as the impression is easily given by the title that the observed reduction in physical activity are a consequence of the pandemic and COVID, where really it is a consequence not of the pandemic or COVID, but rather a consequence of quarantine measures! I therefore suggest something along the lines of the following: ‘Reductions in physical activity level in patients with NMD is a consequence of quarantine during COVID’

R: We strongly agree with the reviewer. We modified the title underlining the after-effects of quarantine instead of the pandemic itself.

  • The second issue is the conclusion: I would conclude that ‘Physical activity levels had returned to normal three years after quarantine, but governments should be careful in implementing quarantine as it has a negative impact on physical activity that in turn has negative consequences for health’. I rather think these messages should be send out, as the way COVID was dealt was highly inappropriate - that is my opinion - but irrespective of my opinion your findings indicate that those quarantine measures had bad consequences for physical activity and that should be the conclusion.

R: Thank you for this comment. We appreciate this suggestion. For this reason, we reviewed and expanded the conclusion following the indications.

  • In the abstract the abbreviations PCS and MCS are used, but they are not explained in the abstract.

R: Thank you for the suggestion. The abstract section has been revised.

  • Of the 9 patients with walking impairment, did they also have a walking impairment at baseline?

R: among the 9 patients with walking impairtment at follow-up, 7 patients presented already walking impairment at baseline. Two patients, both affected by CIDP, presented walking worsening during pandemic. 

  • On page 12 you use the word ‘difference’, but where possible, please rather tell whether it was increased or decreased.

R: Thank you for this comment. The term has been corrected in the manuscript.

  • Is the y-axis in figure 2 really ‘delta MET’? I think it is just ‘MET’.

R: Thank you for this comment. We have inserted “deltaMET” to highlight the variation of this parameter.

  • Page 7 line 275: I am not sure that ‘the maximum increase in walking MET confirms the role of daily walking for a healthier lifestyle’ It is PART of that lifestyle, but not confirmation. It reads like: ‘’Walking is good for you. More walking we have seen. This confirms walking is good for you’.

R: Thank you for the suggestion. We changed the manuscript according to the indications.

  • It seems you have also pre-pandemic data. Would it not be interesting then to see if the PA levels, PCS and MSC etc had returned to pre-pandemic levels, or remained lower?

R: In this study, we decided to explore the topic of quarantine through a comparison between quarantine and post-pandemic. However, we consider this comment very interesting, We will evaluate future studies to deepen this issue.

  • Page 9 line 351-352: Your study did NOT introduce a new training method. You perhaps discussed it in the Discussion, but you did not introduce it in the Methods, or described any results of the training in the Results section. This, therefore, is an inappropriate comment that should be removed.

R: We have included suggestions for the execution of a new training methodology to suggest a practical application to our study. Telecoaching training could help patients with neuromuscular diseases to increase the time spent on physical activity, resulting in an effective, safe, and risk-free training method.

  • Page 9 line 358: Mortality was NOT a major issue in your study. It was just 3% (and perhaps an expected percentage for these conditions in 3 years. A larger problem was drop out (5%) and even more a lack of traceability (26%) (see page 4 for these numbers).

R: We thank the reviewer for this interesting comment. We agree and we have already included this as a main limitation of the study. Probably the high rate of drop out may also be related to the fact that patients affected by neuromuscular disease are used to change their physicians and they are no longer followed by the same clinician for long periods (i.e., 3 years). This is particularly true for orphan diseases (such as Hereditary Spastic Paraplegia, muscular dystrophies, Charcot-Marie-Tooth disease, etc…), as these patients are interested in new experimental trials and are disposed to travel in order to start a new treatment. However, there are no longitudinal data on this cohort of patients and we believe that they are precious, also because the quarantine and pandemic are not reproducible events. Hence, this might be the only opportunity to learn from this unprecedent situation after 3 years in NMD.

  • The implication is that one should be extremely hesitant to introduce another lock-down/quarantine period!

R: Thank you for this comment, we appreciate this suggestion. We have included this consideration in our manuscript.

Hoping in a positive feedback we look forward to hearing from you soon.

Kind regards,

Vincenzo Di Stefano

Round 2

Reviewer 1 Report

Comments and Suggestions for Authors

The reviewer would like to thank the authors for the careful revisions. The current version is certainly easier to read, and it provides interesting and helpful recommendations for future care. There are only a handful of suggested changes, listed below.

L51-53 - Suggest: "Charcot-Marie-Tooth disease is the most common...". Delete the previous sentence, "In addition, within neuromuscular pathologies..."

L99 - Suggest, "...that is, after returning to pre-pandemic levels, in patients..."

L109 - Suggest, "To be included in the research protocol, all participants..."

L116 - I believe this is an error - it is "up to three years since the start of the pandemic..." - not since the end of it.

L142 - Suggest, "...allows measurement...". Same line, delete "mentor"

L300 - Suggest, "...that reduce..." instead of "reducing"

L308 - Suggest, "...correlation" instead of "connection"

L343 - Suggest: "...strategies and avoiding quarantine, if possible."

L350 - The study was conducted as a follow-up, 3 years after the start of the pandemic.

L360-1 - Delete comma before the full stop. Suggest: "However, a significant number of patients opted not to participate in the follow-up survey." - or perhaps something similar.

Comments on the Quality of English Language

Minor editing is needed - I have indicated some of the areas of concern.

Author Response

We thank the reviewer for these precious comments and the possibility of improving the manuscript. We revised the manuscript according to the provided suggestions and we double-checked it for grammar and mistakes. Here we attached the revised version of the manuscript for possible publication in JCM.

Thank you again,

Kind regards,

V. Di Stefano
